# Inhaled Short-Acting Beta Agonist Treatment-Associated Supraventricular Tachycardia in Children: Still a Matter of Concern in Pediatric Emergency Departments?

**DOI:** 10.3390/children10040699

**Published:** 2023-04-08

**Authors:** Bertrand Tchana, Carlo Caffarelli

**Affiliations:** 1Pediatric Cardiology Division, Parma General and University Hospital, 43126 Parma, Italy; 2Clinica Pediatrica, Azienda Ospedaliero-Universitaria, Department of Medicine and Surgery, University of Parma, 43126 Parma, Italy

**Keywords:** supraventricular tachycardia, asthma, bronchospasm, β-2 agonists

## Abstract

Inhaled selective short-acting β-2 agonists (SABA), such as salbutamol, are the rescue treatment of choice for the relief of symptoms of acute asthma exacerbations: one of the leading causes of pediatric emergency department admission and hospitalization. Cardiovascular events, including supraventricular arrhythmias, are the most frequent side effects reported with inhaled SABA in children with asthma and are the main reason for a continuing debate about their safety, despite their widespread use. Although supraventricular tachycardia (SVT) is the most common potentially serious dysrhythmia in children, the incidence and risk factor of SVT after SABA administration is currently unknown. We here reported three cases and conducted a review of the literature in an attempt to gain insight into this issue.

## 1. Introduction

In children, acute exacerbations of asthma are frequent and represent one of the main reasons for emergency department (ED) visits and hospitalization [1]. Inhaled selective short-acting β-2 agonists (SABA), such as salbutamol, are recommended as the rescue treatment of choice for the relief of acute symptoms and prevention of exercise-induced bronchospasm [2,3]. Cardiovascular events are the most frequent side effects reported with inhaled SABA in children with asthma. They include tachycardia, supraventricular arrhythmias, and diastolic hypotension [2,3]. Thus, despite their widespread use, the safety of SABA has been questioned and is still debated because of these cardiovascular effects and the potential arrhythmogenic issues they drive. Higher doses of inhaled SABA treatment may increase the incidence of arrhythmias such as supraventricular tachycardia (SVT) [4]. In children, SVT represents the most frequent and potentially serious dysrhthmia [5,6,7,8]. In this population, the causes of SVT include congenital heart disease, Wolff–Parkinson–White syndrome, drugs (most often sympathomimetic medications), and fever. In infants, 50% of cases are classified as idiopathic. Patients with SVT present with variable clinical features, depending on the duration of the dysrhythmia, ranging from tachycardia alone to congestive heart failure with acidosis and shock [9,10]. The incidence and risk factors of developing SVT after treatment with SABA are currently unknown. Although SABA is a selective β-2 agonist, significant cardiac arrhythmias may result from their use; in a study with salbutamol in 83 children, no significant arrhythmias were triggered [11], while other studies showed that fenoterol increased the heart rate more than salbutamol and terbutaline [12,13]. β-2 agonists increase the rate of conduction across the AV node [14], and their use may lower the threshold for reentry and trigger episodes of SVT. Since asthma is a common condition with salbutamol administration, this rare but significant complication is a matter of concern among pediatricians. Here, we report on three pediatric patients (Table 1) who presented with a potentially life-threatening SVT during salbutamol treatment and reviewed the literature to assess the dimension of the problem, as well as the possible issues and real impact on the management of children with asthma.

## 2. Case Reports

### 2.1. Patient 1

An 11-year-old female with a medical history of allergic asthma since the age of 6 years was brought to our emergency room with her parents after complaining of tachycardia, agitation, and tremors, especially in her hands. Besides long-term treatment with fluticasone and montelukast, she received 2.5 mg nebulized salbutamol when needed, and if necessary, repeated doses were given every 20 min for a total of 3 doses, then up to every 4 h. A physical examination, which was otherwise normal, was remarkable for tachycardia, with a heart rate of 160 beats per minute. Pulse oximetry saturation was 98%. Electrocardiography (EKG) showed a regular tachycardic rhythm with a narrow QRS complex and an apparent absence of P waves. After the failure of a brief trial of Valsalva maneuvers, intravenous adenosine was administered, resulting in the termination of the SVT and the appearance of an EKG with a short PR interval with “delta wave” as per ventricular pre-excitation, which was diagnostic for an AV-reentry tachycardia. An electrophysiological study was carried out and later confirmed the presence of an anterograde unidirectional right posterior AV accessory pathway with characteristics of poor conductive capacity.

### 2.2. Patient 2

A 2-year-old girl with a history of paroxysmal SVT (Figure 1), for which she was treated with flecainide, presented in the emergency room for an episode of acute wheezing. She was treated with inhaled steroids and salbutamol with the resolution of her respiratory symptoms. She was afterward diagnosed with allergic asthma and started on long-term treatment with fluticasone spray, 100 mcg twice a day, and salbutamol as needed, in the case of bronchospasm or asthma exacerbation (1.25 mg), which was eventually repeated every 20 min for a total of 3 doses, and then up to every 4 h. During the follow-up, she suffered one episode of paroxysmal tachycardia, which resolved spontaneously; she also presented with an exacerbation of asthma for which intensive treatment with nebulized salbutamol was administered, 1.75 mg/dose every 20 min for three times, then 1.25 mg every 30 min three times, without the recurrence of SVT.

### 2.3. Patient 3

A 10-year-old male was visited at our emergency room because of tachycardia which had occurred for about two hours. On general practitioner indications, he was on treatment with oral antibiotics, steroids, and inhaled salbutamol for the febrile illness of the upper airways and was last administered with salbutamol before the onset of tachycardia. His history, otherwise unremarkable, revealed, in the previous 4 months, recurrent brief episodes of tachycardia, which resolved spontaneously. Upon arrival, he was placed on a monitor, and his initial vital signs were as follows: heart rate, 225 beats per minute; respiratory rate, 30 per minute; blood pressure, 110/60 mmHg; oxygen saturation, 100% on room air. The physical examination was normal except for a regular tachycardic pulse. EKG showed a regular tachycardic rhythm with a narrow QRS complex and the apparent absence of P waves. During the first assessment, the heart rhythm converted spontaneously to a normal sinus rhythm with an HR of 120 bpm. A few minutes later, an SVT appeared, and after a brief trial of Valsalva maneuvers, an intravenous (IV) catheter was placed into his left ante cubital fossa, and a dose of 3 mg of adenosine was given by rapid IV push followed by a rapid IV flush of saline with conversion to sinus rhythm (Figure 2). A few minutes later, he presented a resumption of tachycardia, which resolved spontaneously.

## 3. Discussion

Tachycardia is a matter of concern during treatment with SABA in children. We reported three cases of children presenting with SVT from non-continuous nebulized salbutamol. We conducted a systematic search aimed at collecting trials and case reports published from 1 January 1991 to 31 December 2022 concerning SVT after non-continuous nebulized/inhaled SABA in children. Six databases were used: Medline, Embase, Cochrane Central, PubMed, Web of Knowledge, and International Pharmaceutical Abstracts. We retrieved five manuscripts [15,16,17,18,19], which described eleven cases in children aged 19 months [16] to 8 years [17] with seven males. Seven of the eleven cases were reported in a single-center hospital-based study that estimated an SVT incidence of 3.9 per 10,000 episodes in SABA treatment and 5.1 per 10,000 asthmatic children treated at a hospital [19]. So far, a small number of cases have been published on this issue. Our study is the second that describes more than one case. It extends knowledge on the characteristics of children presenting with this uncommon adverse effect of inhaled SABA. At variance from previous reports, we found that SVT also occurred in older children. We did not observe a male predominance in accordance with the only case series published until now [19].

The reason for SVT being associated with inhaled SABA that should be taken into consideration is that the atrial and ventricular myocardium consists of both β1- and β2-adrenergic receptors [20,21,22,23,24]. Although β1-adrenoceptors predominate, β2-adrenergic receptors represent 20 to 40% of the total number of β-receptors in the human heart, with a higher density in the nodal tissue compared to the surrounding myocardium and a higher density in the atrioventricular node compared to the sinus node [25,26,27]. The functional responses mediated by β2-adrenoceptors are not necessarily different from those mediated by β1-adrenoceptors. β2-Adrenoceptors are also present on the adrenergic nerve terminals in the heart, where they facilitate noradrenaline release and may contribute to the cardiac effects of β2-adrenoceptor agonists [28,29]. A study of adults with SABA administrated by infusion demonstrated that β2-adrenergic stimulation produces the shortening of the action potential duration in the nodes and in atrial and ventricular muscle, an increase in upstroke velocity on the action potential in the AV node but no effect on the action potential in the His-Purkinje system [13,14]. Tachycardia is due either to reflex cardiac stimulation from peripheral vasodilation, which reduces venous return, resulting in the activation of sympathetic nervous system reflexes and increased cardiotonic and chronotropic effects [14,26,30], or the direct activation of the sinoatrial, right atrial and left ventricular β2-adrenoceptors, which can trigger SVT. Salbutamol produces significant changes in cardiac electrophysiologic properties and significantly shortens the sinus cycle length and sinus node recovery time, enhancing AV nodal conduction and decreasing atrial and ventricular refractoriness, which predisposes to and facilitates the induction of re-entrant and triggered arrhythmias [14,26,28,30,31,32,33]. The QT interval represents the duration of the entire process of depolarization and repolarization of the ventricular myocardium. QTd is the difference between the maximum and minimum QT interval when measured from a 12 leads ECG and describes the heterogeneity of ventricular repolarization; an increase in QTd has been shown to increase the risk of serious arrhythmias [34]. A significant increase in QTc after the inhalation of albuterol and a significant increase in QTd after the inhalation of fenoterol has been observed [35]. Other important triggers of SVT induced by SABA are electrolyte abnormalities, which mainly alter potassium serum balance. SABA can produce increased potassium entry into the skeletal muscle, which may lead to hypokalemia [31,36], especially with the use of corticosteroids. Hypokalemia usually decreases conduction velocity and, based on its severity, may lead either to repolarization and slowdown in the myocardial tissue resulting in increased refractoriness, or the shortening of the refractoriness, triggering arrhythmias. β2-agonists were also shown to cause hypoxaemia by decreasing PaO_2_ and by increasing the blood flow through poorly ventilated areas of the lung and thereby increasing ventilation/perfusion mismatch and hypoxaemia, which can also enhance the risk of rhythm disturbances [26,30,31,36]. In children, a percentage of inhaled SABA was deposited in the oropharynx, swallowed, and systemically absorbed, determining their side effects. Therefore, there are different mechanisms through which inhaled SABA may affect the cardiovascular system inappropriately, producing side effects expressed mainly in rhythm disturbance. In post-marketing and regulatory agencies, reports of these cardiovascular side effects, except for tachycardia, are described as rare (arrhythmias include atrial fibrillation and supraventricular tachycardia, hypokalaemia, peripheral vasodilatation, and hypotension). In a study to assess the degree of safety in the use of inhaled SABA, sinus tachycardia occurred in 7% at a standard dosage and 17% at a higher dosage, without any SVT or QTc prolongation [37]. This study may be in accordance with other data linking the incidence of tachycardia to SABA dosage. In another retrospective study, cohorts of patients were compared between the ages of 2 and 21 when treated in pediatric emergency departments for acute asthma exacerbation before and after the implementation of a high dose of continuous nebulized albuterol protocol. Despite the limits of the study design, no significant adverse effects, including tachyarrhythmia and symptomatic hypokalemia, were found [34]. On the other hand, a single-center retrospective study over a ten-year period found seven cases of SVT in children [19]. Overall, B2-adrenergic receptor activity is the result of a complex mechanism of development, maturation, activation, desensitization, and down-regulation, in which genetic polymorphism, but also selectivity, affinity, and efficacy of the β2-adrenergic agonists [38,39,40,41] play an important role. One can speculate that SVT after inhaled SABA is rare in children, and without substantial morbidity or mortality, because of the state of the development and maturation of the β2-receptor in children and the absence of clinical conditions that altered the distribution of β-receptors in the heart. Another explanation may be that oxidative stress is increased during asthma exacerbations [42,43], which may favor the onset of tachyarrhythmias, and this is more common in adults [44].

Another question is whether there are conditions favoring the onset of SVT after inhaled SABA treatment in children. Importantly, we observed that congenital heart disease and prior arrhythmias were predisposing factors for SVT episodes and were triggered by inhaled salbutamol. This is partially in agreement with previous findings showing that in 11 children with SVT associated with inhaled SABA, five cases had no heart disease [15,16,17,19], four had a positive history of SVT [19]: one of heart disease [19] and one of the Wolff–Parkinson–White syndrome (WPW) [18].

SABA can elicit SVT associated with ventricular pre-excitation. It has been shown that β-agonists enhance the antegrade and retrograde conduction of the accessory pathway in patients with a WPW syndrome, potentially resulting in SVT [38]. In adults with WPW, there were reports of the repeated administration of inhaled SABA without the induction of SVT [39]. In almost all the cases, the patients had been treated with SABA without other episodes of SVT. Some clinical cardiovascular conditions, mainly those leading to heart failure, which is frequent in adults, induce downregulation of β1-receptors, increasing β2-receptor numbers [40].

Finally, in previous studies on SVT associated with SABA in children, conversion was obtained by adenosine in eight children [15,17,18,19], by synchronized electrical cardioversion in a child who did not respond to adenosine [19], and by facial ice [16] or spontaneous recover [20] in one child, respectively. In agreement with previous studies in infants and children [15,17,18,19], we found that SABA-induced SVT is efficiently and safely treated with adenosine which should be considered the first-line treatment for SVT in hemodynamically stable patients, even if they fail to respond to vagal maneuvers. Adenosine is fast acting with transient and limited side effects, and it may potentially help to define the mechanism of tachycardia.

## 4. Conclusions

SVT, after inhaling SABA at the recommended clinical doses, is rare in children, even at larger doses such as those used in asthmatic exacerbations, and does not lead to substantial morbidity or mortality, in accordance with the scarcity of published case reports. However, these numbers may be influenced by several factors, such as non-published single cases or asthma diagnostic criteria, falsely inflating or deflating the estimation of the incidence. The safety of SABA in children with prior cardiac conditions (congenital heart diseases, supraventricular arrhythmias) may be a matter of concern, but in an inpatient setting, SABA-induced SVT can be safely and effectively managed. Thus, more studies with a larger number of cases are needed to establish the real incidence of SVT during treatment with SABA in children.

## Figures and Tables

**Figure 1 children-10-00699-f001:**
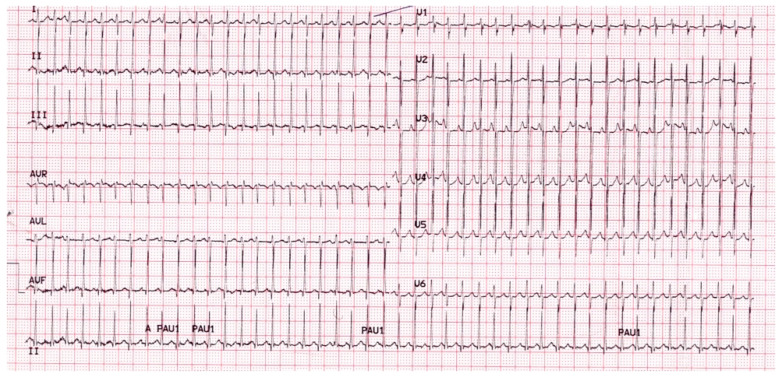
Twelve leads ECG of the patient with paroxysmal supraventricular tachycardia.

**Figure 2 children-10-00699-f002:**
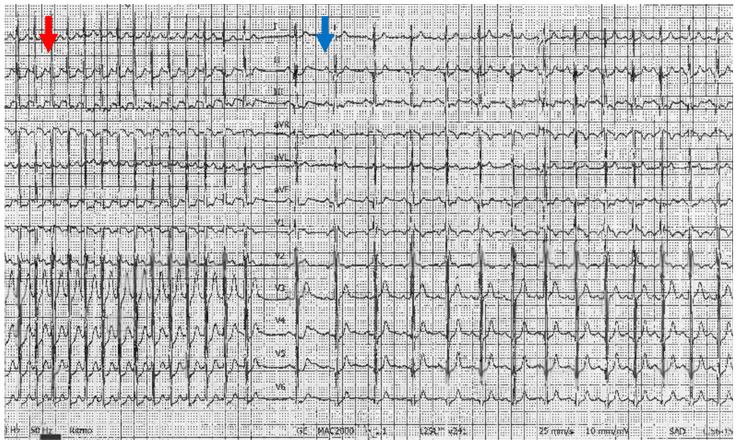
Twelve leads ECG of the patient, featuring a paroxysmal supraventricular tachycardia and the conversion to sinus rhythm (blue arrow) after IV administration of adenosine (red arrow).

**Table 1 children-10-00699-t001:** Features of patients with supraventricular tachycardia (SVT).

	Patient 1	Patient 2	Patient 3
Age (years)	11	2	10
Gender	F	F	M
History	Allergic Asthma	Arrhythmias PSVT Allergic Asthma	Recurrent tachycardia
Dosage of salbutamol eliciting SVT	2.5 mg by nebulizer	100 mcg by pMDI with spacer	300 mcg by pMDI with spacer
Conversion	Adenosine e.v.	Spontaneously	Spontaneously

pMDI, pressurized metered dose inhaler.

## Data Availability

The data presented in this study are available on request from the corresponding author. The data are not publicly available due to privacy or ethical restrictions.

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
