# Peer review of "Inhaled Short-Acting Beta Agonist Treatment-Associated Supraventricular Tachycardia in Children: Still a Matter of Concern in Pediatric Emergency Departments?"

_children, 2023, doi:10.3390/children10040699_

Round 1
Reviewer 1 Report
Dear authors,
I have now completed the review of the manuscript titled "Inhaled Short-Acting Beta Agonist treatment associated Supraventricular Tachycardia in children: still a matter of concern in pediatric emergency department?"
In the present study, the authors report on three cases and conduced a review of the literature, trying to have a dimension of the problem.
The manuscript is interesting and, in general, fair written.
I have some suggestions to further improve the quality of the manuscript.
1. The introduction section introduced some relevant articles. Please explain the results or summarize with effect sizes.
2. I suggest authors clarify how other researchers can obtain the original data.
3. Since there is no clear answer yet as to whether there is a causal relationship between SABA and SVT, authors should discuss about its underlying pathway. For example, SABA can affect heart conductors and cause SVT.(Ref)
Reference: https://thorax.bmj.com/content/64/9/739
4. Although SABA is a widely used drug to treat obstructive respiratory disease, but only some cases like in this report have reported that SVT incidence may increase after SABA inhalation. It implies this association does not apply equally to all patients and may vary depending on individual risk factors or underlying diseases, which should be suggested in this manuscript.(Ref)
Reference: https://www.tandfonline.com/doi/full/10.1080/02770903.2019.1709867
5. What is the future scope of the proposed research, authors have described the limitations in a good way, and I suggest that these can be the future scope of the work.
Author Response
We review the paper according to the reviewer's remarks.
1. in the introduction ee wanted to present the problem by reminding the impact of asthma acute exacerbation on the emergency department, the incidence of supraventricular tachycardia in children, the role of beta-2-agonist in the management and treatment of asthma, and recalling some studies highlighting cardiovascular side effects of the SABA.
2. The original data can be obtained directly from the authors.
3. In the discussion, we analyzed the studies and data in the literature on how SABA may affect the conduction system and eventually cause SVT.
4. The activity of Beta-2 agonist receptors is the result of a complex mechanism involving the development, maturation, activation, desensitization, and down-regulation of the receptors; genetic polymorphism seems to play an important role and may partially explain the difficulties in identifying risk factors excepted for some underlying diseases.
Reviewer 2 Report
This manuscript describes a case report in which 3 pediatric patients with asthma were investigated about their supraventricular tachycardia (SVT) after short acting β-2 agonist (SABA) use, and clinical data were collected. The authors found that most of the patients can be recovered by proper treatment, and the overall SVT-related mortality and morbidity were low. This study focuses on the occurrence and outcome of the SABA-therapy related SVT in children with asthma, which brings some scientific significance. I appreciate that the reported cases included both situations with and without arrhythmia history. I listed several major concerns need to be addressed.
1. Check the English language errors and improper use in the text, including but not limited to:
line 12 “choice for relief of symptoms of acute exacerbations of asthma one of the leading causes” to “choice for the relief of symptoms of acute asthma exacerbation, one of the leading causes”;
line 15 “main raison” to “main reason”;
line 16 “If supraventricular tachycardia (SVT) is the most common” to “Although supraventricular tachycardia (SVT) is the most common”;
line 18 “We report on three cases and conduced a review of the literature, trying to have a dimension of the problem” to 18 “We here reported three cases and conduced a review of the literature, trying to gain an insight into this issue”;
line 203 “their number may be influence “ to “their number may be influenced“.
Please do extensive polishing to the language and correct the errors.
2. Could the authors support the onset EKG of the SVT in the reported cases as a figure in the main text?
3. Do the authors do or plan to do any follow-up to the reported patient? I’m curious about the long-term recurrence of the induced SVT.
4. Please offer an ethical statement or permission from your ethic committee.
Author Response
1. We received the reviewer's suggestions on language issues and provided an extensive revision.
2. We provided an image of the supraventricular tachycardia upon arrival in the emergency room.
3. The patients are in regular follow-ups in the pediatric cardiology division for their arrhythmia:
- as reported the first patient underwent an electrophysiologic study and as far as the accessory pathway presented poor conductive capacity and was close to the His bundle the electrophysiologist decided not to proceed to the ablation all are in treatment, she is on oral prophylaxis with flecainide, and up to now she didn't any recurrence of SVT
- the second case is still on flecainide, she didn't have any recurrence of supraventricular tachycardia, her ECG is normal without any sign of ventricular pre-excitation or other anomalies; because of her young age, and also because of her parent's choice she has not been proposed for electrophysiologic study yet. In the future, particularly if she wants to start a competitive sport, she will undergo electrophysiologic study.
- the third case is in therapy with flecainide; During a period he was complaining of a sensation of tachycardia, and an ECG monitoring with an external loop recorder show that in three weeks ha only two brief episodes of SVT lasting less than one minute.
4. Permission from the ethic committee and formal consent from the parents were obtained for the publication of these cases.
Reviewer 3 Report
An interesting and educational manuscript that has clinical merit. However, there are some editing issues that the authors should consider and address. The following are suggestions/comments regarding those issues. Line 12, "... acute exacerbations of asthma, one of the leading ...". Line 46, "presented with a potentially life-threatening ...". Line 47, "literature to assess the dimension of the problem, possible ...". Line 52, "... our emergency room with her parents after complaining of tachycardia, ...". Line 57, "with a heart rate of 160 ...". Line 59, "... Valsalva maneuvers, intravenous ...". Lines 66 & 67, "... for which she was treated with ...". Line 68, "... with resolution of her respiratory symptoms." Line 69, "... asthma and started on a long-term ...". Line 70, "... fluticasone spray, 100 mcg twice a day, and salbutamol ...". Line 71, "... asthma exacerbation (1.25 mg), eventually ...". Line 73, "... she also presented with an exacerbation of ...". Lines 80 & 81, "... airways and had last administered salbutamol for ...". Lines 82 & 83, "... episodes of tachycardia, which resolved spontaneously." Line 87, "... rhythm with a narrow QRS complex ...". Lines 89 & 90, "... trial of Valsalva maneuvers, an intravenous (IV) ...". Line 107, "... children treated at a hospital (23)." Lines 109 & 110, "... characteristics of children presenting with this uncommon ...". Line 118, "... sinus node (29,30,31). The functional ...". Line 122, "... agonists (32,33). A study of adults with SABA ...". Line 129, "... direct activation of the sinoatrial, right ...". Line 130, "B2-adrenoceptors, which can trigger SVT." Line 149, "In children, a percentage of inhaled ...". Line 153, "... regulatory agencies report these cardiovascular ...". Line 158, "... prolongation (40). This study may be in ...". Lines 172 & 173, "... during asthma exacerbations (46, 47) may favour the onset of tachyarrhythmias, which is more common ...". Lines 188 & 189, "... increasing B2-receptor numbers (44)." Line 193, "...recover (23) in one child, respectively." Line 203, "However, these numbers may be influenced by several ...".
Author Response
We received the editing suggestions of the reviewer and provided an extensive revision of the language.